# Clearing up Clear Cell: Clarifying the Immuno-Oncology Treatment Landscape for Metastatic Clear Cell RCC

**DOI:** 10.3390/cancers13164140

**Published:** 2021-08-17

**Authors:** Sai Krishnaraya Doppalapudi, Zev R. Leopold, Akshay Thaper, Alain Kaldany, Kevin Chua, Hiren V. Patel, Arnav Srivastava, Eric A. Singer

**Affiliations:** Section of Urologic Oncology, Rutgers Cancer Institute of New Jersey and Rutgers Robert Wood Johnson Medical School, New Brunswick, NJ 08901, USA; zrl7@rwjms.rutgers.edu (Z.R.L.); thaper@rwjms.rutgers.edu (A.T.); alain.kaldany@rutgers.edu (A.K.); kc1133@rwjms.rutgers.edu (K.C.); hiren.patel4@rutgers.edu (H.V.P.); srivasar@rwjms.rutgers.edu (A.S.)

**Keywords:** renal cell carcinoma, immunotherapy, kidney cancer, immune checkpoint inhibitors

## Abstract

**Simple Summary:**

Renal cell carcinoma (RCC) is the most common cancer of the kidney. Historically, patients with disease extending beyond the kidney had poor oncologic outcomes. However, discovery of the underlying causes of RCC and the creation of therapeutic agents to target these pathways has revolutionized the management of metastatic RCC. Application of these agents alone or in combination with surgery continue to show encouraging results. In this review, we explore the clinical trial landscape of metastatic RCC, specifically the clear cell type, with particular emphasis on current and upcoming trials utilizing immunotherapeutic agents.

**Abstract:**

Patients with advanced or malignant renal cell carcinoma at the time of diagnosis have historically had a poor prognosis. Immunonologic agents have significantly altered the therapeutic landscape and clinical outcomes of these patients. In this review, we highlight recent and upcoming clinical trials investigating the role of immunotherapies in clear cell RCC. In particular, we emphasize immunotherapy-based combinations, including immune checkpoint inhibitor (ICI) combinations, neoadjuvant, and adjuvant ICI, and ICI agents combined with anti-VEGF therapy.

## 1. Introduction

Worldwide, renal cell carcinoma (RCC) accounts for nearly 5% of all cancers in males and 3% of all cancers in females. Among these incident RCCs, clear cell histology (ccRCC) comprises 75%, the most common subtype [1]. The rising incidence of RCC is often attributed to the increased use of axial imaging in patients with nonspecific gastrointestinal complaints. This has led to the increased incidental detection of small renal masses. However, an important proportion of detected disease is advanced in nature with up to 17% of patients having metastatic disease at the time of diagnosis. In addition, between 20 and 40% of patients with localized disease who initially underwent extirpative surgery will eventually develop distant metastasis [2].

Patients with metastatic RCC (mRCC) can be categorized into various risk groups based on well-established clinical and laboratory factors. Data consolidated from clinical trials during the interferon α (IFN-α) era of systemic therapy led to the development of the Memorial Sloan-Kettering Cancer Center (MSKCC) prognostic model (Table 1). As treatment of mRCC was revolutionized by targeted therapy against the vascular endothelial growth factor (VEGF) pathway, a more contemporary prognostic model was developed by the International mRCC Database Consortium (IMDC). Patients with 0 risk factors are designated as favorable risk, those with one to two risk factors—intermediate risk, and those with >3 risk factors—high risk [3,4].

The recent adoption of immune checkpoint inhibitors (ICIs) has led to an upgraded arsenal to combat mRCC. While immunotherapy with IL-2 and IFN-a maintains an important historical presence and still represents a valuable therapeutic option in the management of mRCC, the present role of immunotherapy in the setting of ICIs represents a paradigm shift. CheckMate-025, reported in 2015, was the first phase III clinical trial to delineate the efficacy of immunotherapy in the setting of mRCC. In this study, advanced and metastatic RCC patients previously treated with antiangiogenic targeted therapy (TT), were randomized to either nivolumab, a PD-1 ICI, or everolimus, an mTOR inhibitor. Those who received nivolumab had a higher median OS (25 vs. 19.6 months, HR 0.73, 98.5% CI 0.57–0.93, *p* = 0.002) and fewer grade 3 or 4 adverse events than those receiving everolimus [5]. Given this encouraging data, interest in utilizing ICI as first line therapy emerged. 

Contemporary clinical trials have adopted combination therapy as the definitive strategy for the management of mRCC. This approach is based on the promising efficacy of combination therapy in multiple different cancer types [6]. There are currently seven phase III randomized controlled trials (RCT) that have adopted either ICI + ICI or ICI + TT as combination therapy in the first line for the treatment of mRCC with a clear cell component (Table 2). The objective of this review is to provide a solid framework to understand the rationale behind combination immunotherapy, to thoroughly discuss the six aforementioned RCTs, and to discuss future directions that should be considered given the ever-burgeoning role of combination immunotherapy in the management of clear cell mRCC. Data regarding non-clear cell histology is beyond the scope of this review. 

## 2. Rationale for Treatment

The treatment paradigm for metastatic RCC has undergone a rapid transformation over the last 20 years. Until 2005, the therapeutic cytokines, high dose IL-2 and IFN-α, were considered the standard of care [7,8]. These early forms of immunotherapy triggered anti-tumor effects by activating cytotoxic T cells and upregulating various other cytokines. The high doses needed to impart effective responses, however, often resulted in significant toxicity requiring intensive care unit level care and in some cases mortality [9,10].

To understand the rationale behind the use of more contemporary targeted therapies and immune checkpoint inhibitors, it is vital to understand the role each pathway plays in tumorigenesis and tumor persistence. In 60 to 90% of sporadic cases of RCC, the von-Hippel Landau (*vhl*) tumor suppressor gene is inactivated via somatic mutation or promotor methylation [11]. This leads to the constitutive activation of hypoxia-inducible factor (*hif*) and subsequent upregulation of HIF-related proteins, most notably VEGF [12]. Another important pathway that ultimately leads to the upregulation of HIF and VEGF is the dysregulation of the PI3K/Akt pathway that leads to activation of mTOR kinase [13]. VEGF ultimately promotes angiogenesis leading to tumor formation, but also induces tumor persistence by promoting the expansion of inhibitory immune cells including regulatory T cells, tumor associated macrophages, and myeloid-derived suppressor cells and by inhibiting the maturation of antigen presenting dendritic cells. In addition, VEGF also promotes PD-1 expression on CD8+ T cells and contributes to their exhaustion by inducing FAS ligand expression on endothelial cells [14]. This mechanism of action of VEGF on the immune system is represented in Figure 1. Contemporary combination immunotherapy trials have utilized a multitude of agents that target this pathway. TTs that were utilized in the six prominent combination immunotherapy RCTs are represented in Figure 2, based on the portion of the VEGF pathway they inhibit.

While targeted therapy has revolutionized the management of mRCC, development of tumor resistance and lack of complete responses has led to the development of a novel treatment strategy known an immune checkpoint inhibition (ICI). When the programmed death receptor 1 (PD-1) receptor on cytotoxic T cells binds to either programmed death receptor ligand 1 or 2 (PD-L1 or PD-L2), the immune cell is inactivated helping the tumor cell evade attack [15]. While PD-1/PD-L1 attenuates T-cell receptor signaling during later stages of the T cell response, CTLA-4, another important player in this pathway, acts as an immune system attenuator in earlier stages, primarily acting on CD4+ effector T cells [16]. Thus, negating these two immune suppressive systems is crucial in preventing tumor persistence. ICIs that were implemented in the six prominent combination immunotherapy RCTs are represented in Figure 3 based on the portion of the ICI pathway they act upon. 

Perhaps the simplest rationale for combining drug interventions in cancer therapy is to simultaneously inhibit multiple pathways to minimize tumor escape. There is, however, more complexity when choosing drug combinations and often one agent can synergistically enhance the effectiveness of the other. Additive toxicity profiles can also make doublet regimens difficult to tolerate, especially in pre-treated populations. Pro-angiogenic factors, such as VEGF, cause the dysregulated formation of leaky tumor vasculature; TTs downregulate this continuous angiogenic signaling causing formation of more mature vessels and, thus, enhancing the local concentration of various systemic agents, most notably ICIs [17,18,19]. Thus, one rational for the combination of VEGF inhibitors and ICIs is their synergistic anti-cancer effect. 

## 3. ICI + TT Combination Therapy

Several phase 3 trials have incorporated the use of one immune checkpoint inhibitor and one targeting agent of the VEGF pathway in the treatment of clear cell mRCC. There are currently five prominent clinical trials with robust data showing the efficacy and safety of this combination strategy (Table 2). Inclusion criteria are shown in Table 3. 

Based on phase 1b trial data demonstrating the antitumor activity of combination pembrolizumab and axitinib in untreated clear cell mRCC, Keynote 426 was developed to compare outcomes in a RCT setting [20]. In this phase 3 trial, patients were randomized in a 1:1 ratio to receive either pembrolizumab 200 mg intravenously every 3 weeks and axitinib 5 mg orally twice daily (AXI-PEMBRO), or sunitinib 50 mg daily on a 4|2 schedule (SUN) [21]. Recently, extended follow-up data have been published from this trial [21]. Overall, 861 patients were randomized with roughly 70% having IMDC intermediate or high risk disease. In the ITT population, patients who received AXI-PEMBRO had improved overall survival (not reached vs. 35.7 months, HR 0.68, 95% CI [0.55–0.85]), improved progression free survival (15.4 vs. 11.1 months, HR 0.71, 95% CI [0.60–0.84]), and higher objective response rates (60% vs. 40%) than those who received SUN. The same findings remained when a sub analysis was performed in patients with intermediate and high risk disease. Overall survival, however, was not superior in patients receiving AXI-PEMBRO in the favorable risk group which may be secondary to the indolent nature of favorable disease and its greater susceptibility to VEGF pathway manipulation. Overall, the total percentage of patients experiencing grade 3 or 4 adverse events was similar between the AXI-PEMBRO and SUN treatment groups (66% vs. 61%). The most common grade 3 or 4 treatment related side effect in the AXI-PEMBRO group was hypertension (22%), increase in ALT (13%), and diarrhea (10%). Interestingly, after adjusting for dose exposure, the number of adverse events of any grade was lower in the AXI-PEMBRO group (63 events per 100 person-months vs. 97 events per 100 person-months). Treatment related side effects lead to the interruption of the axitinib and pembrolizumab combination in 30% of patients and a complete discontinuation in 7% of patients (although higher numbers experienced interruption or discontinuation of one of the two drugs in the combination). In the sunitinib group, drug interruption was experienced in 44% of patients and discontinuation in 12% of patients. Based on this data, AXI-PEMBRO has been approved by the FDA as first line treatment for patients with metastatic RCC [22].

Phase 1b data from a single group, nonrandomized trial with 55 advanced RCC patients undergoing therapy with axitinib and avelumab boasted objective response rates of 58% at 52 weeks of follow-up. A higher rate of objective responses appeared to be associated to the presence of at least 1% PD-L1 expression on tumor-associated immune cells [23]. This data prompted the development of the phase 3 trial, JAVELIN Renal 101, which randomized patients to either avelumab 10mg/kg intravenously every 2 weeks and axitinib 5 mg orally twice daily (AXI-AVE) or sunitinib 50 mg orally on a 4|2 schedule (SUN) [24]. Patients who received AXI-AVE had improved progression free survival regardless of PD-L1 status when compared to those who received SUN (13.3 vs. 8.0 months, HR 0.69, 95% CI [0.56–0.84]). In addition, patients who had PD-L1 positive tumors received a marginally higher relative progression free survival than the overall population when compared to sunitinib (13.8 vs. 7.0 months, HR 0.62, 95% CI [0.49–0.78]). AXI-AVE also sported higher anti-tumor activity in both patients with PD-L1 positive tumors (55.2% vs. 25.5%) and the overall population (51.4% vs. 25.7%). Overall survival data at this point is still immature, however, with a HR of 0.828 (95% CI [0.6–1.15]) in the PD-L1+ population and a HR of 0.796 (95% CI [0.62–1.03]) in the overall population [25]. AXI-AVE and SUN had roughly equal safety profiles with the regimens causing grade 3 or higher adverse events in 71.2% and 71.5% of patients, respectively. The most common grade 3 or higher adverse events reported in the AXI-AVE group were hypertension (25.6%), increased ALT (6.0%), and palmar-plantar erythrodysesthesia syndrome (5.8%). Treatment related side effects resulted in the discontinuation of both avelumab and axitinib in only 7.6% of patients while 13.4% of patients discontinued sunitinib secondary to toxicity. AXI-AVE is currently approved by the FDA for frontline use in patients with advanced RCC [22]. 

Another combination strategy was utilized in IMmotion151 which explored dual inhibition with simultaneous atezolizumab and bevacizumab. Interestingly, bevacizumab was used historically in combination with IFN-α, an older form of immunotherapy, but its use fell out of favor due considerable patient morbidity and lack of sustained durable benefit [26,27]. In this phase 3 trial, patients were randomized in a 1:1 ratio to either atezolizumab 1200 mg and bevacizumab 15 mg/kg intravenously every 3 weeks (ATEZO-BEV), or the standard of care sunitinib 50 mg orally daily on a 4|2 schedule [28]. Median progression free survival favored combination therapy in patients who received ATEZO-BEV in both the ITT (11.2 vs. 8.4 months, HR 0.83, 95% CI [0.70–0.97]) and the PD-L1 positive population (11.2 vs. 7.7 months, HR 0.74, 95% CI [0.57–0.96]) when compared to those receiving SUN. However, overall survival differences did not reach significance in the initial analysis. Of note, the ATEZO-BEV regimen appears to have particular efficacy in RCC with sarcomatoid features, a variant with a generally poor prognosis. In a subset analysis, patients receiving ATEZO-BEV had longer progression free survival regardless of PD-L1 expression (8.3 vs. 5.3 months, HR 0.52, 95% CI [0.34–0.79]) and achieved impressive objective response rates (49% vs. 14%) including complete responses in 10% of patients as opposed to 3% with SUN [29]. ATEZO-BEV was associated with fewer grade 3 or 4 adverse events (40% vs. 54%), lower discontinuation rate secondary to treatment related side effects (5% vs. 8%), and a longer drug exposure period prior to symptom interference (11.3 vs. 4.3 months, HR 0.56, 95% CI 0.46-0.68) when compared to SUN. Importantly, patients receiving ATEZO-BEV reported milder treatment and disease related symptoms associated with a decreased quality of life compared to those receiving SUN. ATEZO-BEV has been approved for unresectable hepatocellular carcinoma, but not for advanced RCC to date. 

Following two phase 3 trials showing improved survival with the use of cabozantinib and nivolumab individually when compared to sunitinib therapy, the checkmate 9ER trial was designed, in which combination cabozantinib and nivolumab (NIVO-CABO) was investigated [5,30]. In this study, patients were randomized in to either nivolumab 240 mg every 2 weeks intravenously and cabozantinib 40 mg orally daily, or to sunitinib 50 mg orally daily on a 4|2 schedule [31]. Patients who received NIVO-CABO had longer progression free survival (16.6 vs. 8.3 months, HR 0.51, 95% CI [0.41–0.64]), overall survival (not reached for both arms, HR 0.60, 95% CI [0.40–0.89]), and higher objective response rates (55.7% vs. 27.1%) with 8% having complete responses compared to 4% in patients who received SUN. As a common theme, NIVO-CABO has also not been shown to improve overall survival in IMDC favorable risk patients (HR 0.84, 95% CI 0.35–1.97). Combination therapy with cabozantinib and nivolumab did cause more grade 3 or 4 adverse events than sunitinib monotherapy (60.6% vs. 50.9%) with hypertension, diarrhea, and various lab abnormalities (hyponatremia, hypophosphatemia, and AST/ALT abnormalities) being most frequently experienced. 19.1% of patients required high dose corticosteroids (≥40 mg of prednisone or equivalent) for treatment related adverse events. In addition, 19.7% of patients discontinued at least one of the two agents due to side effects and only 5.6% discontinued both; sunitinib, in contrast, was discontinued in 16.9% of patients due to adverse events. Importantly, patient reported metrics (FKSI-19 questionnaire) showed that although scores were similar between both arms, quality of life was maintained overtime with NIVO-CABO, while a gradual deterioration was noted with SUN. NIVO-CABO is currently approved by the FDA for the frontline treatment of metastatic RCC [22].

The CLEAR trial, the most recent trial investigating combination immunotherapy in metastatic RCC, is unique in that patients were assigned to three different treatment arms. Lenvatinib and pembrolizumab as monotherapies have been previously shown to have efficacy against advanced RCC and in a phase 1b/2 trial their combined use has also shown promising antitumor activity [32,33]. In this study, patients were randomized 1:1:1 to either lenvatinib 20 mg orally daily and pembrolizumab 200 mg every 3 weeks intravenously (LEN-PEMBRO), lenvatinib 20 mg orally daily and everolimus 5 mg orally (LEN-EVERO) daily, or to sunitinib 50 mg orally daily on a 4|2 schedule (SUN) [34]. Patients on LEN-PEMBRO had longer OS (NR vs. NR, HR 0.66, 95% CI [0.49–0.88]) and PFS (23.9 vs. 9.2 months, HR 0.39, 95% CI [0.32–0.49]), higher objective response rates (71% vs. 36.1%) and complete response rates (16.1% vs. 4.2%), and a longer duration of response (25.8 vs. 14.6 months). LEN-EVERO also sported improved PFS compared to SUN (14.7 vs. 9.2 months, HR 0.65, 95% CI [0.53–0.80]), but unlike LEN-PEMBRO did not show improved OS (NR vs. NR, HR 1.15, 95% CI [0.88–1.50]). Notably, PFS was improved with LEN-PEMBRO when compared to SUN amongst all MSKCC and IMDC risk groups; in addition, expression of PD-L1 did not impact the effectiveness of this treatment regimen. Grade 3 or 4 adverse events occurred in 82.4% of patients who received LEN-PEMBRO, 83.1% of patients who received LEN-EVERO, and 71.8% of patients who received SUN. The most common treatment related side effects that occurred in all three groups was diarrhea, hypertension, an elevated lipase level, and hypertriglyceridemia. Notably, both LEN-PEMBRO and LEN-EVERO had higher rates of proteinuria than SUN, and SUN had higher rates of palmar-plantar erythrodysesthesia syndrome. In the LEN-PEMBRO group 13.4% of patients had to discontinue both medications and 78.4% had to interrupt at least one of the medications secondary to treatment related adverse events. In the LEN-EVERO group, 18.9% discontinued both drugs, and 73.2% had dose interruptions of at least one drug under similar circumstances. Drug interruptions in these regimens did not differ very significantly from the SUN regimen, which experienced a 14.4% discontinuation rate and a 53.8% drug interruption rate. Due to the tolerability and efficacy of the LEN-PEMBRO regimen, it is currently receiving priority review by the FDA for the frontline treatment of metastatic RCC [35]. 

## 4. ICI + ICI Combination Therapy

Given the success of immune checkpoint inhibitors in CheckMate 025 in the second line setting after treatment with antiangiogenic therapy, a paradigm shift has occurred propelling these agents to the first line for treatment naïve mRCC. In a phase 2 study, ipilimumab monotherapy at a dose of 3 mg/kg was associated with a 12.5% objective response rate but was associated with significant immune-mediated toxicity with 43% of patients having grade 3 to 5 adverse events [36]. Given that combination therapy with both nivolumab and ipilimumab has shown enhanced tumor activity in a variety of cancers, CheckMate 016, a phase 1 trial, sought to determine the optimal regimen for safety in patients with mRCC. Patients who received nivolumab (3 mg/kg) and ipilimumab (1 mg/kg) had equivalent objective response rates at 40.4% and significantly lower grade 3 or 4 adverse events (38.3% vs. 61.7%) than those who received nivolumab (1 mg/kg) and ipilimumab (3 mg/kg) [37]. It appears that high dose ipilimumab is the most prominent factor when establishing toxicity in this patient population. Thus, nivolumab (3 mg/kg) and ipilimumab (1mg/kg) became the dose regimen utilized in the phase 3 trial CheckMate 214. 

CheckMate 214 has produced provocative data favoring the use of combination ICI therapy over the previous standard of care, sunitinib monotherapy, in patients with previously untreated mRCC with a clear cell component. In this trial, patients were randomized in a 1:1 ratio to either nivolumab (3 mg/kg) and ipilimumab (1 mg/kg) every 3 weeks for 4 doses followed by nivolumab (3 mg/kg) every 2 weeks (IPI-NIVO), or sunitinib 50 mg orally once a day for 4 weeks followed by 2 weeks off therapy, a 4|2 schedule (SUN) [38]. Recently, 4-year durable follow-up data have been published. In total, 1096 patients were randomized with an intention to treat (ITT), with approximately 80% of this population having IMDC intermediate or high risk disease. In the ITT group, patients who received IPI-NIVO had improved overall survival (not reached vs. 38.4 months, HR 0.69, 95% CI [0.59–0.81]) and higher objective response rates (39.1% vs. 32.4%) than those who received SUN. When analyzing patients with intermediate or high risk disease, those who received IPI-NIVO had improved overall survival (48.1 vs. 26.6 months, HR 0.65, 95% CI [0.54–0.78]), improved progression free survival (11.2 vs. 8.3 months, HR 0.74, 95% CI [0.62–0.88]), and higher objective response rates (41.9% vs. 26.8%) when compared to those who received SUN. Of note, improved overall survival occurred in this group regardless of PD-L1 expression, although a more pronounced effect was seen in those with ≥1% expression when compared to those with <1% expression (HR 0.45 vs. 0.79). 

Interestingly, in the favorable risk population, while OS benefit for either treatment arm was inconclusive with a 4-year minimum follow up, IPI-NIVO underperformed SUN in both progression free survival (28.9 vs. 12.4 months, HR 1.84, 95% CI [1.29–2.62] and objective response rates (29.6% vs. 51.6%). This comes with a caveat, however, as patients who received IPI-NIVO had nearly double the complete response rate (12% vs. 6.5%) in this population. In addition, favorable risk patients are more likely to achieve these survival benefits off therapy when originally receiving IPI-NIVO, thus avoiding treatment related toxicity. In terms of safety, patients in the IPI-NIVO group had fewer grade 3 or 4 adverse events (47.9% vs. 64.1%) than those in the SUN group. The most common grade 3 or 4 toxicities in the IPI-NIVO group were elevated lipase levels (10%), fatigue (4%), and diarrhea (4%). In the SUN group the most common grade 3 or 4 toxicities were hypertension (16%), palmar-plantar erythrodysesthesia (9%), and fatigue (9%). Treatment related adverse events lead to the discontinuation of a patient’s drug regimen in 22.7% in the IPI-NIVO arm and in 13.1% in the SUN arm. Finally, 29.1% of patients treated with IPI-NIVO received high dose glucocorticoids (≥40 mg of prednisone daily) for any grade adverse events [38,39]. Given the favorable profile of this data, the Food and Drug Administration (FDA) has approved combination therapy with nivolumab and ipilimumab for the treatment of patients with treatment naïve mRCC with intermediate or high risk disease.

## 5. Cytoreductive Nephrectomy (CN) + ICI

Historically, CN, with the addition of systemic cytokines, was the standard of care for patients with metastatic RCC. Supportive data were purported by two major randomized trials, SWOG 8949 and EORTC 30947, which showed a significant survival advantage in patients who received CN and cytokine therapy versus those who received cytokine monotherapy [40]. The role of CN recently came into question after the publication of the CARMENA trial results, which indicated that sunitinib monotherapy was non-inferior to CN followed by sunitinib [41]. However, this trial was contaminated by a significant proportion of patients who were in the high risk category or had a large volume of metastatic burden outside the kidney. In addition, the slow accrual witnessed in this trial highlights a potential recruitment bias in which lower risk patients chose not to enroll [42]. The SURTIME trial further investigated the role of CN by exploring the ideal order of systemic therapy and extirpative surgery. While this study suffered from poor accrual, the ITT analysis showed a significant overall survival advantage (32.4 vs. 15 months, HR 0.57) in patients who received systemic targeted therapy prior to CN [43]. Ultimately, there is still a prominent role of CN in the advanced RCC population, but patient selection is of utmost importance.

With the advent of combination immunotherapy, the role of CN is being reinvestigated. In an NCDB study by Singla et al., 391 surgical candidates diagnosed with treatment naïve mRCC received either CN and ICI or ICI alone. With a median follow-up of 14.7 months, patients who underwent CN had improved overall survival (NR vs. 11.6 months, HR 0.23). Those who received ICI prior to CN had decreased pathologic T stages, grades, tumor sizes, and lymphovascular invasion rates than those who had upfront CN. In addition, 10% (2 of 20 patients who underwent CN after ICI) had complete responses and pT0 disease on post-operative pathology [44]. This large retrospective cohort study demonstrated a profound role of CN in the era of combination immunotherapy and made way for investigative clinical trials. 

Numerous clinical trials have burgeoned from the idea that deferred CN is the optimal approach for patients with mRCC who are surgical candidates (Table 4). Given the potent anti-tumor effects of immune checkpoint inhibition, deferred CN allows patients to receive the benefit of systemic therapy while avoiding treatment delay and ineffective extirpative surgery in cases of disease progression. NORDIC-SUN (NCT03977571) and PROBE (NCT04510597) are two examples of phase 3 clinical trials that seek to investigate whether CN and perioperative combination immunotherapy potentiates an overall survival benefit compared to combination immunotherapy alone in patients with advanced RCC [45,46]. In addition, CYTO-KIK (NCT04322955) is a notable phase 2 trial whose primary goal is to determine the number of mRCC patients who will achieve complete response following treatment with perioperative nivolumab and cabozantinib [44]. Of note, each of the aforementioned trials has a histological component with the goal of identifying biomarkers that can predict response to combination immunotherapy. While the introduction of novel immune system based systemic therapies vastly outpaces clinical trial data, the results of these aforementioned studies will be crucial to determine the future role of CN in this era of advanced RCC treatment.

## 6. Stereotactic Radiation (SR) + ICI

Traditionally, RCC has been thought to be resistant to conventional fractionations of SR. However, it is now known that high doses per fraction can overwhelm the natural radio-resistance of RCC and provide an effective means of local therapy. In a meta-analysis of 28 studies which included mRCC patients with oligometastatic disease, 90% of lesions treated with SR did not show radiographic progression at 1 year and only minimal toxicity was noted (~1%) [47]. In a more recent retrospective analysis, 60 patients with histologically confirmed RCC (88.3% with ccRCC) and radiographic evidence of visceral or lymph node metastasis who underwent robotic radiosurgery, a variant of SR, were shown to have excellent local tumor control (96.7%) and minimal adverse events (8.3%) with only one grade 4 AE noted [48]. It is thus conceivable that SR provides an effective alternative for cytoreduction in patients with metastatic disease, and in combination with combined immunotherapy, may provide a survival benefit especially in poor surgical candidates. 

There are now several ongoing clinical trials involving SR and immunotherapy that are showing promise. The basis of these studies is from preclinical data suggesting that SR releases tumor antigens thus working synergistically with ICIs. For instance, early data from the phase 2 NIVES trial showed improved response rates to nivolumab in irradiated metastatic sites (26.9%) when compared to non-irradiated sites (17.4%) supporting the aforementioned preclinical findings [49]. In addition, RADVAX RCC (NCT03065179) and CYTOSHRINK (NCT04090710) are phase 2 randomized trials, which seek to evaluate objective responses to SR and combination ipilimumab and nivolumab in patients with oligometastatic RCC [49,50]. Results of these later two trials are especially important as they involve patients with IMDC intermediate or high risk disease who are not surgical candidates. Thus, if efficacy is demonstrated, a cytoreductive option is available for this specific patient population.

## 7. Triple Therapy Regimens

Given the success of two agent combination therapies, the natural progression would be to implement a third agent, which preferably acts upon a different portion of the tumorigenesis pathway and to study this new combination’s safety and efficacy. Adopting triple therapy regimens begets two important questions, however: (1) how do we adjust the dose of each individual agent in order to optimize safety and efficacy? (2) Will the targeting of multiple pathways in the first line reduce the efficacy of second line therapies in the cases of disease progression? There is currently one active phase 3 trial investigating such regimens involving the use of a HIF-2α antagonist, and there are numerous phase 1 and 2 trials underway utilizing a plethora of different agents (Table 5).

Although combination immunotherapy has incurred significant survival advantage and durable response rates in patients with advanced RCC, many will inevitably experience disease progression thus necessitating targeting another portion of the tumorigenesis pathway. When the *vhl* tumor suppressor gene, which plays an important role is RCC pathogenesis, is inactivated, an accumulation of HIF-2α occurs, which subsequently dimerizes HIF-1β leading to the upregulation of oncogenic proteins, such as VEGFA, PDGFB, TGFα, cyclin D1, and CXCR4 [51]. Targeting HIF will thus undoubtedly play a crucial role in the treatment of mRCC. 

A dose escalation phase 1 trial which studied the first-generation HIF-2α antagonist, PT2385, in patients with previously treated advanced ccRCC showed complete and partial response rates of 2% and 12% respectively with satisfactory tolerability [52]. These results propagated the development of a second-generation HIF-2α inhibitor, MK6482. Results from a phase 1/2 trial and phase 2 trial involving MK6482 were recently presented at GU ASCO. In one study, 55 heavily pretreated patients with mRCC (81% received prior PD1/PD-L1 inhibitors and 92% received prior VEGF inhibitors) who were then treated with MK6482 achieved a 25% objective response rate and a median PFS of 14.5 months [53]. Another investigation which studied combination cabozantinib and MK6482 showed an ORR of 20% and 90.2% experienced tumor shrinkage in previously treated patients [54]. In addition, MK6482 appears to be generally well tolerated with the most common grade 3 or 4 AEs being anemia (27%) and hypoxia (16%), which were readily treatable with erythropoietin stimulating agents and supplemental oxygen. In addition, no new side effects were noted when MK6482 was implemented in combination with cabozantinib. There are currently several phase 3 comparative trials underway utilizing MK6482 in combination with TKIs in the second-line setting after ICI failure [55,56]. In fact, impressive preliminary data have already launched HIF-2α agents into first line treatment regimens with combination immunotherapy. An early phase 3 trial will be recruiting patients to determine if the addition of MK6482 to LEN-PEMBRO, the regimen utilized in the CLEAR trial, will provide an improved survival advantage [57].

In addition, included in other triple therapy phase I/II regimens are various novel agents with unique drug targets, such as bempegaldesleukin (an engineered Il-2 cytokine prodrug), IPI-549 (a PI3K-ϒ inhibitor acting upstream of mTOR), CBM588 (a strain of Clostridium butyricum which may alter the gut microbiome and improve ICI effectiveness), and entinostat (a histone deacetylase inhibitor with synergistic antitumor effects with immunotherapy such as high dose Il-2) [58,59,60,61].

## 8. Adjuvant Immunotherapy for High-Risk Localized ccRCC

Nearly 20–40% of patients with localized RCC will develop distant metastasis even after surgical extirpation [62]. In fact, those patients in the highest risk categories have nearly a 50% chance of recurring within 6 years of nephrectomy [63]. Multiple predictive factors have been identified to stratify patients into various risk categories. These factors include ECOG performance status, nuclear tumor or Fuhrman grade, tumor stage, nodal involvement, margin status, microvascular invasion, necrosis, and sarcomatoid or rhabdoid differentiation [64]. Various risk stratification calculators such as the MSKCC, UISS, Leibovich, and SSIGN nomograms have been designed to identify patients with the highest probability of developing recurrence and metastasis as these patients will theoretically attain the highest benefit from adjuvant therapy regimens. Fundamentally, the objective of administering adjuvant therapy is to eliminate micrometastatic disease that could not be debulked from primary extirpation. This strategy has been employed in a number of other types of solid tumors, and thus numerous clinical trials have been opened for patients with RCC in order to identify an effective regimen. Given the robust data surrounding combination immunotherapy in the metastatic setting for patients with ccRCC, it makes logical sense that such regimens may also provide benefit in the adjuvant setting as well. 

Interim data from adjuvant treatment with TKIs and mTOR inhibitors have to this point not shown consistent survival benefit in the adjuvant setting. In fact, only one clinical trial, S-TRAC, was able to show disease free survival benefit, resulting in a rather half-hearted FDA approval of sunitinib after nephrectomy in high-risk patients [65]. One theory that has been suggested as to why only S-TRAC was able to show survival benefit was that it implemented the strictest inclusion criteria, only allowing patients with at least pT3 disease to participate. ASSURE (sunitinib vs. sorafenib vs. placebo), PROTECT (pazopanib), ARISER (girentuximab), ATLAS (axitinib), and SORCE (sorafenib for 3 years vs. sorafenib for 1 year then placebo for 2 years) all allowed patients with high grade pT1b disease and pT2 disease thus potentially including a subset of patients that may be less likely to harbor micrometastatic disease and would thus contaminate survival results for high-risk patients [66,67,68,69,70]. In a post-hoc analysis of the ASSURE trial in which only the highest risk subset of patients was analyzed (pT3, pT4, or N+), no survival benefit was noted despite more stringent characterization of this high-risk population [10]. These results may at least be partially attributable to the fact that nearly 20% of the ASSURE treatment arm included patients with non-clear cell histology, as disease free survival benefit was seen on the ATLAS subset analysis. Nevertheless, the general consensus at this time is to not offer adjuvant therapy in high-risk localized disease outside of a clinical trial. 

Given the impressive data surrounding ICIs and the durable responses they provide especially in ccRCC, several clinical trials have been developed to study their efficacy in the adjuvant setting. Checkpoint inhibition may provide an improved response to micrometastatic disease as it instills direct cytotoxic effects rather than acting secondarily by preventing angiogenesis, which may not be a critical component of micrometastasis tumor biology. General trial design includes administration of therapeutic agents 8 to 12 weeks after surgical extirpation of clear cell or sarcomatoid differentiated RCC. Interim results of IMMotion 010 (atezolizumab), Keynote 564 (pembrolizumab), Checkmate 914 (ipilimumab and nivolumab), PROSPER (neoadjuvant and adjuvant nivolumab), and RAMPART (durvalumab vs. durvalumab and tremelimumab) are pending [71,72,73,74]. Ultimately, if results from these phase III trials are promising, a variety of combination immunotherapy regimens would likely be studied as is being done in the metastatic setting, providing an exciting frontier in the management of high-risk localized ccRCC. 

## 9. Future Considerations

Combination immunotherapy regimens have become the standard of care in the treatment of mRCC with a clear cell component, given robust data from high quality RCTs. Despite providing an improved survival advantage compared to sunitinib monotherapy, an important question that will arise when devising new trials is determining a proper comparator group. While caution needs to be taken when comparing data from different trials, perhaps future drug regimens should utilize LEN-PEMBRO as the comparator group as patients in this trial achieved the highest progression free survival (23.9 months) in all risk categories, the highest rates of objective response (71.0%) and complete response (16.1%), and the highest percentage of population living at two years (79.2%) [34].

Many patients on combination immunotherapy will also experience serious treatment related immune-related toxicities. The ability to predict which patients will respond most profoundly will allow us to maximize clinical benefit; this idea emphasizes the importance of predictive candidate biomarkers. Unfortunately, no biomarkers to date have proven to be applicable clinically. Perhaps the most logical biomarker, PD-L1, was not shown to accurately determine clinical response to ICIs. Patients throughout the seven prominent RCC discussed had potential for objective and even complete response despite PD-L1 status. This finding may be secondary to the variability of both commercially available PD-L1 clones and heterogenous expression of PD-L1 in various tumor regions [75,76]. Ultimately, the complex immunologic milieu of each individual patient appears to be the most important criteria when determining the efficacy of combination immunotherapy. This may also be why PD-L1 status fails to predict therapeutic effectiveness of immunomodulatory therapy; it is in fact the complex interplay of specific T-lymphocyte subsets, absence of immunosuppressive elements, and the quality of the local immune contexture that are greater determinants [77]. There is much room to grow in this area and numerous studies are being conducted to discover these predictive biomarkers. 

Finally, given the advent of a new era of systemic therapy for clear cell mRCC, it is perhaps an opportune time to revisit the current risk stratification system. It is likely that as the IMDC risk stratification system implemented new risk factors for assessing disease severity, data from contemporary RCTs will add or replace risk factors as well. This new system can be utilized in future RCTs and will be an integral part of trial design.

## 10. Conclusions

Given the impressive survival data seen in contemporary combination immunotherapy trials, the armamentarium for the treatment of advanced clear cell renal cell carcinoma is bound to change swiftly. With immune checkpoint inhibitors at the backbone of future regimens, combination therapies will include cytoreductive surgery, radiation, mechanistically unique targeting agents, and the use of three or more agents in one regimen. With the implementation of such strategies, it will be paramount to balance the risks and benefits of such regimens. Careful patient selection will remain critical and novel biomarkers will be needed to help identify subpopulations most likely to benefit from intensified treatment regimens [78,79].

## Figures and Tables

**Figure 1 cancers-13-04140-f001:**
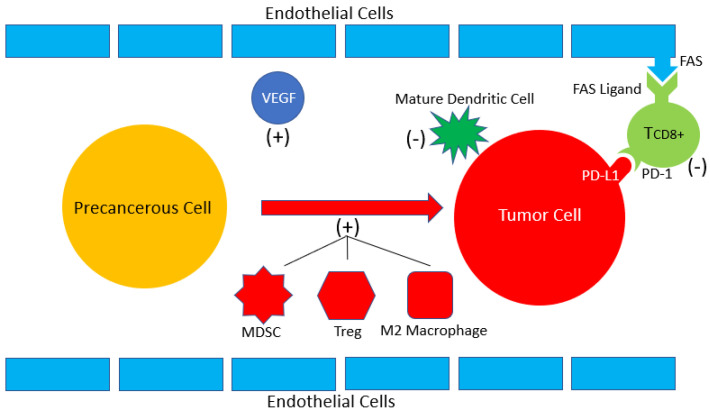
VEGF promotes many immune mediated effects that promotes tumorigenesis. VEGF upregulates immunosuppressive cells such as myeloid-derived suppressor cells (MDSCs), T regulatory cells (Treg), and M2 macrophages which promote tumor cell formation as indicated by the (+) sign. In addition, VEGF prevents the maturation of dendritic cells and reduces the effectiveness and number of CD8+ T cells by inducing PD1/PD-L1 interactions and by promoting FAS binding. Because dendritic cells and CD8+ T cells normally inhibit tumorigenesis, a (-) sign is seen adjacent to their depictions.

**Figure 2 cancers-13-04140-f002:**
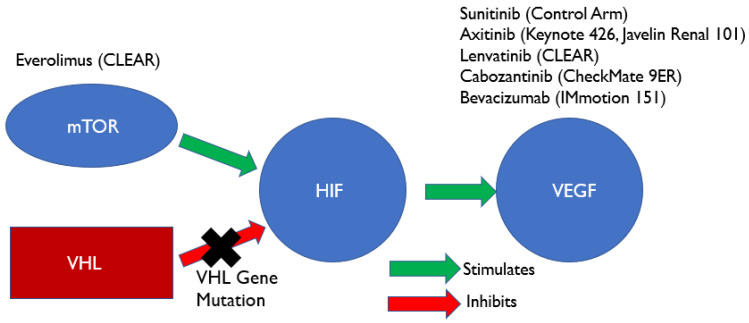
Mechanism of angiogenesis in sporadic RCC. Everolimus–inhibitor of mTOR. Sunitinib–tyrosine kinase inhibitor of VEGF. Axitinib-TKI of VEGF. Lenvatinib–multiple kinase inhibitor of VEGF. Cabozantinib–TKI of RET, MET, and VEGFR2. Bevacizumab–anti-VEGF monoclonal antibody.

**Figure 3 cancers-13-04140-f003:**
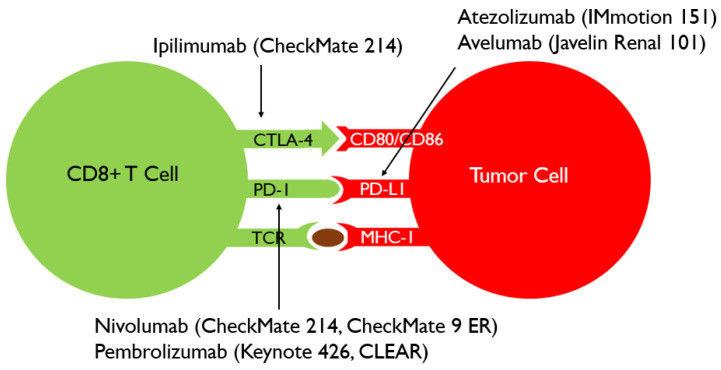
Schema of tumor induced T-cell anergy and immuno-oncology agents. Ipilimumab–anti-CTLA4 monoclonal antibody. Nivolumab–anti-PD-1 monoclonal antibody. Pembrolizumab-anti-PD-1 monoclonal antibody. Atezolizumab-anti-PD-L1 monoclonal antibody. Avelumab-anti-PD-L1 monoclonal antibody.

**Table 1 cancers-13-04140-t001:** Risk factors included in the MSKCC and IMDC prognostic models. Patients with 0 risk factors are designated as favorable risk, one to two risk factors—intermediate risk, and those with ≥3 risk factors—high risk. ✓ included in model. X not included in model.

Variable	Value	MSKCC	IMDC
Karnofsky Score	<80	✓	✓
Diagnosis to treatment time	<12 months	✓	✓
Hemoglobin	<Lower limit of normal	✓	✓
Corrected Calcium	>10 mg/dL	✓	✓
LDH	1.5× normal	✓	X
Platelets	>400k	X	✓
Neutrophils	7 × 10^9^	X	✓

**Table 2 cancers-13-04140-t002:** Phase III clinical trials reporting data in mRCC treated with ICI + ICI or ICI + TT. OS—overall survival. ORR—objective response rate. PFS—progression free survival. NE—not estimable. HR—hazard ratio. CI—confidence interval.

Trial Name (NCT)	Phase	Status	Experimental Arm(s)	Comparator	Primary Outcome(s)	Primary Result(s)
CheckMate 025 (NCT01668784)	III	Active, not recruiting	Nivolumab	everolimus	OS	25.0 months (95%CI 21.8 to NE) vs. 19.6 months (95%CI 17.5–23.1)
CheckMate 214 (NCT02231749)	III	Active, not recruiting	nivolumab + ipilimumab	sunitinib	ORR	42% vs. 27%, *p* < 0.001
					OS	NR vs. 26.0; HR, 0.63, *p* < 0.001
					PFS	11.6 vs. 8.4 months; HR 0.83, *p* = 0.03
CheckMate 9ER (NCT03141177)	III	Active, not recruiting	nivolumab + cabozantinib	sunitinib	PFS	16.6 month (95%CI 12.5–24.9) vs. 8.3 month (95%CI 7.0–9.7)
CLEAR (NCT02811861)	III	Active, not recruiting	lenvatinib + everolimus	sunitinib	PFS	14.7 vs. 9.2 months; HR, 0.65; 95%CI, 0.53–0.80; *p* < 0.001
			lenvatinib + pembrolizumab			23.9 vs. 9.2 months; HR, 0.39; 95%CI, 0.32-0.49; *p* < 0.001
IMmotion151 (NCT02420821)	III	Active, not recruiting	atezolizumab + bevacizumab	sunitinib	PFS (PD-L1+)	11.2 vs. 7.7; HR 0.74, 95%CI 0.57–0.96
					OS (ITT)	HR 0.93, 95%CI 0.76–1.14
Javelin Renal 101 (NCT02684006)	III	Active, not recruiting	avelumab + axitinib	sunitinib	PFS (PD-L1+)	13.8 vs. 7.0; HR 0.62, 95%CI 0.49–0.77, *p* < 0.0001
					OS (PD-L1+)	HR 0.828, 95%CI 0.596–1.151, *p* = 0.1301
Keynote 426 (NCT02853331)	III	Active, not recruiting	pembrolizumab + axitinib	sunitinib	PFS	15.4 vs. 11.1 months, HR 0.71, 95%CI 0.60–0.84, *p* < 0.0001
					OS	NR vs. 35.7 month; HR 0.68, 95%CI 0.55–0.85, *p* = 0.0003

**Table 3 cancers-13-04140-t003:** Inclusion criteria of phase III clinical trials exploring advanced or metastatic RCC. *PROSPER RCC included M1 disease that is planned to be definitively treated within 12 weeks of nephrectomy such that the patient is M0, no evidence of disease.

Trial Name (NCT)	Histology	Stage	Prior Systemic Treatment
CheckMate 025 (NCT01668784)	Clear-cell component	Advanced or Metastatic RCC	One or two prior anti-angiogenic therapies permitted
CheckMate 214 (NCT02231749)	Clear-cell component	Advanced or Metastatic RCC	One prior therapy in resectable RCC permitted except those targeting VEGF
CheckMate 9ER (NCT03141177)	Clear-cell component	Advanced or Metastatic RCC	One prior therapy in resectable RCC permitted except those targeting VEGF
CLEAR (NCT02811861)	Clear-cell component	Advanced or Metastatic RCC	No prior systemic anti-cancer treatments for RCC permitted
IMmotion151 (NCT02420821)	Clear-cell component	Advanced or Metastatic RCC	No prior systemic anti-cancer treatments for RCC permitted
Javelin Renal 101 (NCT02684006)	Clear-cell component	Advanced or Metastatic RCC	No prior systemic anti-cancer treatments for RCC permitted
Keynote 426 (NCT02853331)	Clear-cell component	Advanced or Metastatic RCC	No prior systemic anti-cancer treatments for RCC permitted
NORDIC-SUN (NCT03977571)	All histological subtypes	Metastatic RCC	No prior systemic anti-cancer treatments for RCC permitted
PROSPER RCC (NCT03055013)	All histological subtypes	cT2-4 Nx, cT1-4 N1, M0*	No prior systemic anti-cancer treatments for RCC permitted
SURTIME (NCT01099423)	Clear-cell subtype	Metastatic RCC	No prior systemic anti-cancer treatments for RCC permitted
PROBE (NCT04510597)	All except collecting duct carcinoma	Metastatic RCC	No prior systemic anti-cancer treatments for RCC permitted

**Table 4 cancers-13-04140-t004:** Clinical trials investigating the role of ICI + CN or ICI + SR. OS—overall survival. CRR—complete response rate. PFS—progression free survival. EFS—event-free survival. ORR—objective response rate. HR—hazard ratio. SBRT—stereotactic body radiation therapy.

Trial Name (NCT)	Phase	Status	Experimental Arm(s)	Comparator	Primary Outcome(s)	Primary Result(s)
Cyto-KIK (NCT04322955)	II	Recruiting	cabozantinib + nivolumab + nephrectomy	-	CRR	Ongoing
NORDIC-SUN (NCT03977571)	III	Recruiting	nivolumab + ipilimumab then CN then nivolumab maintenance	nivolumab + ipilimumab then nivolumab	OS	Ongoing
PROSPER RCC (NCT03055013)	III	Recruiting	nivolumab then nephrectomy	Nephrectomy	EFS	Ongoing
SURTIME (NCT01099423)	III	Completed	nephrectomy + sunitinib	sunitinib then nephrectomy then sunitinib	PFS	42% vs. 43%; *p* = 0.61
PROBE (NCT04510597)	III	Recruiting	nivolumab or pembrolizumab or avelumab	(nivolumab or pembrolizumab or avelumab) + CN	OS	Ongoing
NIVES (NCT03469713)	II	Active, not recruiting	nivolumab + SBRT	-	ORR	Ongoing
RADVAX (NCT03065179)	II	Completed	nivolumab + ipilimumab + SBRT	-	ORR	56%; 90%CI 38.7–78.9%

**Table 5 cancers-13-04140-t005:** Clinical trials investigating the combination of three or more systemic therapies in ccRCC.

Trial Number/Name	Phase	Histology	Drug(s)	Comparator Arm	Primary Endpoint	Status
PIVOT IO 011 (NCT04540705)	Phase I	CC component	Nivolumab + bempegaldesleukin + axitinib	-	AEs	Recruiting
	Phase II		Nivolumab + bempegaldesleukin + cabozantinib	Nivolumab + Cabozantinib	ORR	
NCT04518046	Phase I/Ib	CC	Sitravatinib + nivolumab + ipilimumab	-	AEs	Recruiting
NCT04413123	Phase II	nCC	4 cycles of cabozantinib + nivolumab + ipilimumab followed by cabozantinib + nivolumab	-	ORR	Recruiting
MARIO-3 (NCT03961698)	Phase II	CC	ipi-549 + atezolizumab + bevacizumab	-	CR	Recruiting
NCT03829111	Phase I	CC component	CBM588 + nivolumab + ipilimumab	Nivolumab + Ipilimumab	Change in Bifidobacterium composition of stool	Recruiting
NCT03024437	Phase I/II	All	Atezolizumab + entinostat + bevacizumab	-	Dose, ORR	Recruiting
NCT02496208	Phase I	CC	Cabozantinib S-malate + ipilimumab + nivolumab	-	Dose, AEs	Recruiting
MK-6482-012 (NCT04736706)	Phase III	CC	Pembrolizumab + belzutifan + lenvatinib	Pembrolizumab + Lenvatinib	PFS, OS	Recruiting
			Pembrolizumab/quavonlimab + lenvatinib			

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
