# Peer review of "Clearing up Clear Cell: Clarifying the Immuno-Oncology Treatment Landscape for Metastatic Clear Cell RCC"

_cancers, 2021, doi:10.3390/cancers13164140_

Round 1

Reviewer 1 Report

This is a well written summary on the current treatment landscape in clear cell renal cell carcinoma. Immunotherapy has changed the treatment options and improved outcomes of patients significantly. However, some concerns about the manuscript have to be raised.

1.) Paragraph 7 on stereotactic radiation and ICI is a little bit short and seems to miss important points. For instance the treatment of oligometastatic and mixed responses with stereotactic radiation in combination with ICI seems an important upcoming treatment option for patients. Retrospective studies emphasize this role and safety of treatment of metastatic lesions of RCC (e.g. PMID: 33567564) and similar concepts has been evaluated in prospective trials (e.g. DOI: 10.1200/JCO.2020.38.6_suppl.614)

2.) A subheading in line 399 seems to be missing. Please add.

3.) Immunotherapy in an adjuvant setting might be of interest for readers and might be included into the review.

Author Response

Thank you for your insight. 

In reference to your comments:

1) Added information regarding NIVES trial and RADVAXX RCC trial. In addition, implemented sources that were mentioned into this section. 

2) Corrected

3) Added section entitled "Adjuvant Immunotherapy for High-Risk Localized ccRCC"

Reviewer 2 Report

The review of Krishnaraya Doppalapud et al. contextualizing ongoing clinical research of combination therapies including immune checkpoint inhibitors for the treatment of clear cell renal cell carcinoma is very extensive. It demonstrates the broad spectrum of clinical studies and combination therapies as well as the efficacy and quality of these studies. It is important, to summarise the large-scale of these studies.

I have a major concern regarding the structure of the single chapters of this review. It would be appreciated if re-structuring could be considered and that visual representations would be added that underline the described research.

Figure 2 is simple and representative for the mechanism of action of the immune checkpoint inhibitors, but as the study names are given, it would be nice to see also the mechanism of the drugs that are given in combination with the ICIs.

Table 1 is difficult to understand because it does not necessarily show how patients are clustered, but which risk factors are considered in the prognosis models. More than 3 risk factors mean poor risk, but I think better phrasing would be high risk.

The inclusion criteria especially for the phase III studies should be presented clearly, as it is an important factor for the final statistics on the OS and PFS.

Because of the structure of each chapter, the information gets lost in the process of reading, that combinations of ICIs are already approved by the FDA, which is one of the most important aspects, summarising the quality of phase III ongoing research leading to future treatment options becoming available to the patients. In this context, the authors could comment, how the current phase III studies may lead to superior treatment than the one already approved.

The authors should also discuss that the immune landscape of ccRCC is patient dependent and has an impact on the prognosis of patients as well as the efficacy of the given single or combined treatment (DOI:10.1038/nrclinonc.2017.101). This will further underline, why PD-L1 is/ was not the best biomarker in the scenario of ccRCC.

Within the future considerations it sounds very negative that patients will be suffering. The future considerations should directly mention the positive and how the authors envision that future considerations lead to better outcomes than the current ones.

Minor comments:

In line 50 it says YEAR, which does not fit with the sentence.

In line 59 it says 7 in line 63 it says 6.

Please explain the timeframe for the considered conference data in Material and Methods. Now it says 'recent', but an explicit timeframe would be better as it is given for the research in the databases.

Looking for NCT01668784 from Table 2 in the NCT database shows 'active, not recruiting'. The information on 'completed' may be cited.

The abbreviation for CI in Table 2 is not given.

IFN-alpha should be revised to be written concisely in the same style.

Author Response

Thank you for your insight. Below are our responses to your comments. 

Figure 2 mentions the mechanism of action of the drugs in its description. 

Table 1 was adjusted to make it easier to understand. In addition, poor risk was changed to high risk. 

Inclusion criteria for important phase 3 trials was tabulated into a new table entitled Table 3. 

During our discussion of each of the 7 major RCTs, we discuss that these drug regimens are FDA approved (except for the IMmotion study). Would you like us to tabulate this instead?

Citation regarding the importance of immune landscape when determining patient response to therapy was added (added to future considerations section)

Our purpose for the future considerations section was not highlight that patients will be suffering, but to purpose ideas that could better optimize treatment. We thus did not make any whole scale changes to this particular section. 

Minor edits were corrected. 

Round 2

Reviewer 2 Report

The corrections and edits are appreciated and nicely done.
However, minor edits remain needed.

In my opinion, the figures do not add to the review. There is more important information in this review that could be highlightesd through a graphical representation. The authors should consider changing the figures.

Table 4 is now given twice.

Please check once again if all interferon-alpha mentioning are written correctly.

Line 85-86: This part of the sentence is very difficult to read.
Line 92: The gene HIF-1a is mentioned, therefore it should be written in small letters and italic. Please follow the international code.
Line 141-144: This sentence is very difficult to read.
Line 149: Poor or high risk? This is also the case for line 295 and 298.
Line 455: The 'highest' benefit.

Author Response

Thank you for your comments.

Instead of removing the figures that are already present, I added an extra figure (now figure 1) that explains how VEGF contributes to immunomodulatory effects that promote tumorigenesis. I believe this allows the reader to better understand the rationale of why VEGF inhibition along with immune checkpoint inhibition can be used in combination to tackle mRCC. 

Table 4 now changed to table 5 and appropriately changed in the body of the paper. 

Interferon alpha was appropriately corrected.

All other minor edits were corrected. When directly referring to a gene, I changed each word to be lower cased and italicized.